# Can intergovernmental cooperative policies promote water ecology improvement—An analysis based on water quality data from China's general environmental monitoring station

**Yu Ding** *, **Chen Gong**

School of Politics and International Studies, Central China Normal University, Wuhan, Hubei, China

* dingyu@ccnu.edu.cn

## Abstract

To strengthen cooperation among local governmental departments and improve water ecology, China has proposed the river management policy "river chief system + procurator". However, it remains to be verified that intergovernmental cooperative policies contribute to the improvement of China's water ecology. Based on data from 87 national water quality monitoring sites released by the China Environmental Monitoring Station from 2015 to 2022, this paper constructed a multiperiod differences-in-differences model to evaluate the effectiveness of the cooperative governmental policy, the "river chief system + procurator", on the improvement of the water quality of rivers and lakes. The results of the study show that cooperative government policy helps to improve the water environment of rivers and lakes, which means that the implementation of the "river chief system + procurator" policy has significantly improved water quality conditions where implemented. In addition, further analysis revealed that intergovernmental cooperative policies had a limited impact on relevant indicators of river and lake pollutants that are more susceptible to different pollution sources compared to the comprehensive indicator of water quality class. This study helps further the understanding of the effects of cooperative intergovernmental policies and the policy practice of environmental governance in China.

## Introduction

It is a common goal of all countries to protect the ecological environment while maintaining economic development. As the ecological maintenance of rivers and lakes is related not only to national development strategies but also to the health and life of everyone within its society, the sustainable development and protection of water resources are receiving increasing attention from the international community [1]. In 2022, UNESCO released the World Water Development Report 2022, stating that equitable access to safe and clean drinking water and sanitation are distinct human rights. However, water conservation faces multiple difficulties in

**Funding:** This research was funded by National Social Science Foundation Youth Program of China, grant number 23CZZ013(DY) and the Humanities and Social Sciences Youth Foundation, Ministry of Education of the People's Republic of China, grant number 22YJC810001(DY). There was no additional external funding received for this study.

**Competing interests:** The authors have declared that no competing interests exist.

implementation. Especially for developing countries with weak formal institutions of accountability [2], they lack a strong bureaucracy to supervise and control the management and maintenance of water resources by local governments [3]. At the same time, the responsibility of water resource protection is often scattered throughout the legislative, administrative, judicial, and other departments leading to a problem of fragmentation and mutual shuffle, which will further increase the administrative costs of water resources governance. Therefore, the protection of water resources and the improvement of the ecological environment cannot be separated from the top-down policy innovation of the country and the exploration of local governments.

In China, the protection of water resources is also impacted by the above problems. To overcome the lack of management subjects and fragmentation of governance in the process of water resource protection, China tried to improve domestic water ecology through policy innovation. At the end of 2016, the General Office of the State Council of the People's Republic of China issued Opinions on the Comprehensive Implementation of the River Chief System in an attempt to improve the water ecology by clarifying who is responsible for the conservation of rivers and lakes. The essence of the river chief policy lies in centralizing the water environment supervision responsibilities originally scattered among the departments of water conservancy, environmental protection, agriculture, forestry, transportation, fisheries, and oceans into the hands of local party and government leaders and relying on their personal authority and resources to promote the environmental protection of rivers and lakes. By doing this, the central government has tried to overcome the adverse effects of fragmented governance through multisectoral cooperation, and some local governments have implemented the water environment management policy of "river chief system + procurator". The uniqueness of this policy is that for the first time, administrative and judicial forces are combined in the water environment management process, and the originally decentralized water management departments are integrated. In the environmental protection of rivers and lakes, the procuratorial authorities and river chiefs both cooperate with each other, supervise each other, and work together to continuously improve water conservation in their jurisdiction. China's practice and exploration of water environment policy provide a good model for us to observe how developing countries improve water ecology and raise the central question of this paper: Can these management policies improve water ecology? Can the emergence of intergovernmental cooperation policies improve water quality?

By reviewing the literature, we found that studies on the impact of national policies on water ecology still have not reached a consistent conclusion [4]. Some researchers have argued that national policies can improve water quality and achieve the initial effect of water pollution control [5,6]. They argue that national policies can gradually clarify who has responsibility for river management, which can effectively supervise the enterprises within its jurisdiction, publicize information about their water use, and prompt them to take more responsibility for environmental protection [7]. National policies can also further clarify the power and responsibility relationship in the process of river management. This helps local governments integrate resources and mobilize multiple forces to open the last mile of water environment management [8].

Other studies conclude that national policies are negatively influenced by administrative departments and administrative divisions, making it difficult to improve the ecological status of rivers and lakes. The government's fragmented governance weakens the effect of water environmental management, and this negative impact is mainly reflected in the COD and NH3H indicators that reflect the degree of water pollution [9]. At the same time, the unique cross-regional characteristics of river and lake management further weaken the implementation effect of governance policies. Some studies have found that there is a "beggar-thy-neighbor"

phenomenon in the pollution control of trans-provincial rivers. The pollution of rivers at the provincial border is more serious than that within the province [10]. Moreover, the length of the river also has a dissipating effect on the effectiveness of river management. The policy effect of the river chief system decreases significantly when the length of the river exceeds 221 km [11].

However, different research conclusions lead us to question the actual effectiveness of national water management policies. In particular, whether local implementation authorities can promote the improvement of water ecology still needs to be further tested and analyzed. This paper selected water quality data released by China's General Environmental Monitoring Station from 2015–2022 as an observation sample and considered the "river chief system + procurator" policy implemented in some Chinese localities as a "quasinatural experiment". More precisely, the implementation of the "river chief system + procurator" policy provided an ideal quasinatural experimental condition for this paper to observe the impact of government cooperation on water environment governance: for one thing, this policy is exogenous in nature. The policy of "river chief system + procurator" is exogenous to the objective of rivers and lake pollution levels, which ensures that there is no reverse causality between explanatory variables and explained variables in this paper. Secondarily, the implementation of this policy is conducive to strengthening the legal supervision of water ecological governance and effectively cracking down on illegal discharge behaviors within the basin. The cooperative mechanism of administrative and judicial forces can give greater play to the role of local governments and break the negative impact of governance fragmentation. Based on this, this paper adopted a time-varying DID method to analyze the actual effect of the "river chief system + procurator" policy on ecological improvement and its mechanism of action.

## Institutional background and characteristic facts

Compared with the mature model of water management in developed countries, China's water pollution management policy system started late but developed rapidly. The exploratory management of water quality in rivers and lakes by local governments has provided an initial experience for the overall improvement of water ecology in China. In 2007, the cyanobacteria pollution crisis in Taihu Lake broke out in Jiangsu Province. To solve the water pollution problem in the basin, Jiangsu Province began to manage its rivers by appointing administrative officials as "river chiefs" on each river. This direct leadership of the officials led to the full integration of technical and administrative forces, and the water quality of Taihu Lake rapidly improved and reached the national standard by the end of 2008 [12]. After Jiangsu Province's approach to water management achieved these positive results, the central government began to promote the "river chief system" nationwide. At the end of 2016, the General Office of the CPC Central Committee and the General Office of the State Council issued Opinions on the Comprehensive Implementation of the River Chief System, and various regions accelerated the pace of implementing the river chief system. As of early 2019, China had more than 1.23 million river chiefs on board. Among them, there are more than 300,000 fourth-level river chiefs and more than 930,000 village-level river chiefs, extending the organizational system to rural areas and opening up the "last mile" of river management.

After the widespread promotion of the "river chief system", local governments in China have started to explore the upgraded collaboration mechanism of the "river chief system + procurator" to further optimize the effectiveness of river management. Compared with the previous "river chief system", this collaborative mechanism attempts to further enhance the effectiveness of the "river chief system" in the protection of rivers and lakes and the continuous improvement of the water ecological environment by activating the judicial role of the procuratorate. As a useful supplement to administrative water management, the judiciary plays an

important role in the process of water management. In the guidance issued jointly by the river chief offices at all levels and the procuratorial authorities on how to carry out the collaboration mechanism, collaborative leadership, information sharing, case collaboration, joint work, daily contact, and other working methods are gradually shaped [13]. Collaborative leadership means that river chiefs and procurators make plans for key work and cases of water area governance through joint inspections and meetings; information sharing and case collaboration emphasize that both sides should communicate and transfer with each other in terms of case clues, inspection results, and rectification feedback; joint work and daily contact also aim to ensure that this synergistic mechanism can function effectively. Currently, 15 provinces have established a collaborative mechanism of river and lake governance between procuratorial organs and river chief offices (water conservancy departments), giving full play to the legal supervision function of procuratorial organs, ensuring the implementation of various measures and tasks of the river chief system, and promoting the formation of joint efforts to protect the river and lake water resources, water ecology, and the water environment. Since the implementation of the "river chief system + procurator" collaboration mechanism, the focus of policy exploration has been to build the joint force of administrative law enforcement and procuratorial supervision, further integrate the water environment management forces, and realize the overall management of the basin environment. Therefore, can this cooperative mode of water management truly play a role in improving the water ecological environment? In fact, a large number of studies remain on a case-by-case basis to analyze the impact of government cooperation or regional cooperation on water ecology and fail to reveal and test whether the policy itself has a substantial effect on the improvement of water quality. Therefore, this paper adopts a time-varying DID model to reveal the effect of the policy of "river chief system + procurator".

## Theoretical mechanisms and research hypotheses

Before the implementation of the river chief system, China's river and lake governance responsibilities were scattered between multiple administrative departments, showing a pattern of fragmented governance and making it difficult to clarify the responsible party for river and lake management. The river chief system assigns the right to control the pollution of rivers in each area to the corresponding departmental leaders who will act as "river chiefs" [14] and be responsible for the management and protection of the corresponding rivers and lakes. This initiative has contributed to the development of a unified river governance policy, penalty standards, and management norms in river governance. Since its introduction in 2007, its effectiveness has been verified in the practice of water governance in some regions of China [15]. The implementation of the river chief system policy clarified the main party responsible for the management of rivers and lakes, laid a basic institutional foundation for the improvement of the water environment, and became a microcosm of China's environmental policy toward holistic governance [9]. However, experience has shown that environmental governance is different from general public affairs management and often requires multidepartment and multiforce cooperation to achieve better governance results. On the basis of the river chief system, the "river chief system + procurator" linkage policy is designed to take the convergence of administrative law enforcement with criminal justice and procuratorial public interest litigation as a starting point and utilize the supervisory power of the judicial sector to urge the government to implement initiatives related to river management in the process of river water quality management.

In view of the remarkable results achieved by the river chief system, we believe that the "river chief system + procurator" policy implemented in China may also improve the

ecological environment of rivers and lakes [16]. In China, the main indicator for measuring the ecological environment of rivers and lakes is the water quality rating data released by water quality monitoring stations around the world. As an important water environment performance assessment indicator, the water quality rating data reflect the comprehensive water quality of a specific river or lake. At the same time, China's Ministry of Ecology and Environment issued the "Program for Ranking the Water Environment Quality of National Surface Water Assessment Sections in Prefecture-level and Above Cities (Trial)", which clearly stipulates that the ranking of the water environment quality of each city is determined in accordance with the water quality level [17]. The water environment is gradually becoming an important reference indicator for the performance assessment of local governments.

This leads us to propose research Hypothesis 1: The implementation of the policy of "river chief system + procurator" can significantly enhance the water quality of rivers and lakes.

In addition to the indicator of water quality class, the China Environmental Monitoring Center (CEMC) included indicators of water pollutants such as the hydrogen ion concentration index (PH), dissolved oxygen (DO), potassium permanganate (CODMn), and ammonia nitrogen (NH3N) to assess the quality of the water environment. Compared with the comprehensive indicator of water quality class, water pollutant indicators can reflect the sources and categories of pollutants, analyze the shortcomings and deficiencies in the process of water environment management, and provide data to support the Chinese government in taking more targeted measures to manage the water environment. We believe that there are differences in the data structure and practical utility between the water quality class indicators and the specific indicators of water pollutants, which need to be distinguished when setting up the research hypotheses.

Therefore, we propose research Hypothesis 2: the implementation of the policy of "river chief system + procurator" can significantly reduce the pollution level of rivers and lakes.

## Materials and methods

### Measurement model setting

Since the time of implementing the "river chief system + procurator" in various regions is not consistent and even some administrative regions have not yet implemented the policy, this paper used a time-varying DID model. DID (difference-in-difference) is based on a counterfactual framework to assess the change in the observed Factor y in both cases of policy occurrence and nonoccurrence, and it is widely used in public policy evaluation because it can more accurately assess the net effect of policy implementation on individuals. Based on the weekly water quality statistics and real-time water quality monitoring system of the China General Environmental Monitoring Station from 2015 to 2022, this study used the time-varying DID model to test the impact mechanism of the "river chief system + procurator" policy on the water ecological environment in river and lake basins. By referring to the method of Thorsten and Alexey [18], a time-varying DID model was constructed:

$$WQ_{it} = \beta_0 + \beta_1 R_{it} + \beta_2 control_{it} + \mu_t + \gamma_i + \varepsilon_{it} \tag{1}$$

Specifically, WQ refers to the water quality of the water quality monitoring point of country $i$ in year $t$, and it is measured by the water quality class index and four water pollutant indicators, including the hydrogen ion concentration index, dissolved oxygen, potassium permanganate, and ammonia nitrogen. $\gamma_i$ is the individual fixed effect, $\mu_t$ is the time fixed effect, $\varepsilon_{it}$ is the disturbance term, and $control_{it}$ includes population, GDP, GDP per capita, public revenue, public expenditure, the share of the primary sector, the share of the secondary sector, and sewage treatment rate as the control variables. The core explanatory variable $R_{it}$ in this paper is the

**Table 1. Description of surface water grade classification.**

| | |
|---|---|
| I | Mainly applicable to source water, national nature reserves |
| II | Mainly applicable to centralized surface water sources of drinking water level 1 protected areas, rare aquatic habitats, fish and shrimp spawning grounds, larvae and juveniles baiting grounds, etc. |
| III | Mainly applicable to centralized surface water sources of drinking water level 2 protected areas, fish and shrimp overwintering grounds, migratory channels, aquaculture areas, and other fisheries waters and swimming areas |
| IV | Mainly applicable to general industrial water areas and recreational water areas where the human body does not come into direct contact |
| V | Mainly applicable to agricultural water areas and general landscape requirements of the water |

Source: GB 3838–2002 Environmental Quality Standard for Surface Water.

interaction term between the individual dummy variable and the time dummy variable and will be used to identify national water quality monitoring sites where the "river chief system + procurator" policy has been implemented in year t. In addition, the model estimation and parallelism trend tests all used robust standard errors.

## Data sources

**Dependent variables.** Water environment monitoring data (Water_quality, PH, DO, CODMn, and NH3N) from the Chinese General Environmental Monitoring Station website real-time water quality monitoring section of the automatic water quality monitoring weekly report. The monitoring data were recorded for a total of 148 monitoring stations at State control points for water quality from 2007 to 2022, monitoring the scope of the water quality, PH, dissolved oxygen, CODMn, and NH3N. According to China GB 3838–2002, "surface water environmental quality standard", surface water grades from low to high were divided into I, II, III, IV, and V. The lower the water quality level, the better the water body environment (see Table 1). PH, dissolved oxygen, CODMn, and NH3N as the water pollutant evaluation indices were used to reflect the degree of surface water pollution. The lower the concentration of dissolved oxygen is, the higher the concentration of CODMn, NH3N, and the water quality level, indicating that the water body environment is worse. The distribution of the hydrogen ion concentration index was monitored in the range of 6–9 (see Table 2). Since 2018, the Chinese General Environmental Monitoring Station has stopped reporting weekly water quality monitoring reports and instead reported real-time water quality monitoring data. Therefore, this paper crawled real-time water quality monitoring data from the database to compensate for missing water quality data so that the panel data had a balanced distribution. After the data processing, we obtained the monitoring data of 87 water quality monitoring stations from 2015 to 2022.

**Table 2. Description of surface water pollutant indicator.**

| Serial number | Item standard values classification | I | | II | III | IV | V |
|---|---|---|---|---|---|---|---|
| 1 | PH(dimensionless) | | | 6~9 | | | |
| 2 | Dissolve Oxygen ≥ | Saturation rate 90% (or 7.5) | | 6 | 5 | 3 | 2 |
| 3 | CODMn ≤ | 2 | | 4 | 6 | 10 | 15 |
| 4 | NH3N≤ | 0.15 | | 0.5 | 1.0 | 1.5 | 2.0 |

Source: GB 3838–2002 Environmental Quality Standard for Surface Water.

Unit: mg/L.

**Independent variables.** The implementation time of the "river chief system + procurator" was manually checked from the websites of local governments at the city and county levels and recorded in detail. Considering that various forms of cooperation between local river chiefs and procurators have been carried out before formal signing, such as joint river patrol, public interest litigation against illegal acts of polluting rivers and lakes, and joint river enforcement, the policy impact effect may have been realized earlier. Therefore, this paper took the year before the signing time of the interdepartmental water management by local governments at the city and county levels as the base year. In this paper, the water environment before the implementation of the "river chief system + procurator" was taken as the control group, and after the implementation of the "river chief system + procurator" was taken as the treatment group. By constructing dummy variables, the control group was assigned a value of 0, and the treatment group was assigned a value of 1. If the coefficient β1 is negative, it means that the water grades and water pollution level are reduced, which indicates that the cooperation model of "river chief system + procurator" helped improve the water ecological environment of the local rivers and lakes.

**Control variables.** The water quality of rivers is affected not only by public policies but also by other economic and social factors. Therefore, we need to integrate other variables into the model for a comprehensive study when examining the relationship between changes in river and lake water environments and the implementation of the policy of the "river chief system + procurator". Referring to Li and Zhou's approach, we believe that the overall condition of the regional water environment is closely related to the local level of economic development, industrial structure, financial income, and population base. Therefore, the control variables in the regression model to test the effectiveness of the "river chief + prosecutor system" include economic and social variables such as population, GDP per capita, public revenue, public expenditure, annual growth rate of GDP, proportion of primary industry, and proportion of secondary industry. Considering that the capacity of municipal wastewater treatment may also affect the regional water environment, we included it as a variable in the empirical model as well. The control variables were taken from the China Urban Statistical Yearbook, and all data were matched to the cities where the monitoring sites were located.

## Results

### Descriptive statistics

Table 3 shows the descriptive statistics of the variables, including the observed value, mean value, standard deviation, and maximum and minimum values of the variables. Among them, the variables population, public revenue, public expenditure, and GDP per capita are logarithmically treated. The river and lake water quality class variables are categorical variables with values ranging from 1 to 6, and the other variables are continuous variables. To reduce the influence of extreme values, the explained variables were averaged. The mean value of the water quality class was 2.519, the mean value of PH was 7.764, the mean value of dissolved oxygen was 9.488, the mean value of CODMn was 3.681, and the mean value of NH3N was 0.411. The results of these variables show that the data distribution is within the normal range.

### Time-varying DID regression results

This study used stata to analyze the effect of the governmental cooperative water management policy of "river chief system + procurator" on the water grades of rivers and lakes, as well as the effect of this policy on river pollutants. Table 4 shows the results of the time-varying DID model, which indicates that the policy of "river chief system + procurator" can significantly improve the water quality of rivers and lakes, but the implementation of this policy failed to

**Table 3. Descriptive statistics of quantitative variables.**

| quantitative variable | observed value | mean value | standard deviation | min | max |
|---|---|---|---|---|---|
| Water_quality | 655 | 2.519059 | 1.029804 | 1 | 6 |
| PH | 669 | 7.763814 | 0.4933646 | 6.38 | 9.4375 |
| DO | 672 | 9.487729 | 2.03677 | 3.093333 | 18.25 |
| CODMn | 658 | 3.680932 | 2.518388 | 0 | 40.475 |
| NH3N | 664 | 0.4109574 | 0.9944335 | 0.019 | 14.73333 |
| Population | 580 | 630.9208 | 452.7641 | 44 | 3416 |
| ln Population | 580 | 6.224632 | 0.7028452 | 3.78419 | 8.136226 |
| financial revenue | 581 | 4454687 | 8369816 | 194636 | 71700000 |
| ln financial revenue | 581 | 14.55388 | 1.127897 | 12.17889 | 18.08732 |
| fiscal expenditure | 581 | 7131772 | 9946031 | 903911 | 83500000 |
| ln fiscal expenditure | 581 | 15.39315 | 0.764776 | 13.71449 | 18.24054 |
| GDP per capita | 577 | 64366.14 | 36377.58 | 15303 | 180044 |
| ln GDP per capita | 577 | 10.91869 | 0.5584917 | 9.635804 | 12.10096 |
| GDP growth rate | 577 | 6.798024 | 2.931215 | -7.5 | 13 |
| Primary industry ratio | 579 | 10.46684 | 8.487203 | 0.27 | 48.52 |
| Ratio of secondary industry | 579 | 43.24953 | 9.095379 | 11.7 | 73.19 |
| Sewage treatment rate | 543 | 90.61615 | 9.970246 | 26.92 | 100 |

improve the water pollution indicators. In other words, the policy of "river chief system + procurator" has a limited effect on reducing the pollution level of rivers and lakes.

Specifically, the governmental cooperative water management policy of "river chief system + procurator" has a significant effect on improving the water quality class of rivers and optimizing the environment of water bodies ($\beta$ = -0.366, $p < 0.001$), and the conclusion is still valid after adding control variables ($\beta$ = -0.327, $p < 0.05$). The data show that the surface water quality at the water quality monitoring stations improved significantly after the implementation of the cooperative policy between government departments. Meanwhile, the results after adding the control variables find that the improvement of the regional water quality class was closely related to the level of the local economic development ($\beta$ = -0.0499, $p < 0.05$) and the growth of the governmental financial revenue ($\beta$ = -0.584, $p < 0.05$), indicating that the improvement of water ecology in each region cannot be separated from the level of local economic development and local financial support.

To further reveal the intrinsic mechanism by which the governmental cooperative water management policy of the "river chief system + procurator" affects the water quality of rivers, this paper referred to the cases of the "River Chief System + Procurator General" jointly released by the Supreme People's Procuratorate of China and the Ministry of Water Resources every year and sorted out the basic forms of cooperation between river chiefs and procurators. Under the above interdepartmental cooperative water management policy, China's river chiefs and procurators have joined forces to establish an effective joint river and lake management platform through various forms of cooperation, such as "joint river patrols", "public interest litigation", "colocation of offices", "information sharing", and "transfer of clues". In the process of regulating the water quality of rivers and lakes, judicial power has a strong deterrent force for illegal sewage behavior, and the attorney general intervenes in river and lake governance, thus enhancing the deterrent force of the law, effectively combating environmental pollution and strengthening the effect of the governance of rivers and lakes. Additionally, the "river chief system + procurator" policy prompts the procuratorate and administrative organs to form a common goal of safeguarding the interests of the ecological environment and the interests of the

**Table 4. Regression models of time-varying DID.**

| | (1) W_quality | (2) CODMn | (3) DO | (4) PH | (5) NH3N | (1) W_quality | (2) CODMn | (3) DO | (4) PH | (5) NH3N |
|---|---|---|---|---|---|---|---|---|---|---|
| **did** | -0.366*** | 0.0207 | -0.0342 | -0.0193 | 0.033 | -0.327** | 0.14 | 0.057 | 0.0455 | 0.0887 |
| | (0.115) | (0.218) | (0.271) | (0.079) | (0.079) | (0.141) | (0.275) | (0.283) | (0.065) | (0.115) |
| **ln Population** | | | | | | -0.248 | 0.54 | 0.121 | 0.128 | 0.402* |
| | | | | | | (0.314) | (0.694) | (0.671) | (0.181) | (0.227) |
| **ln financial revenue** | | | | | | -0.584** | -1.045 | 0.877 | -0.22 | -0.594* |
| | | | | | | (0.241) | (0.661) | (0.558) | (0.164) | (0.352) |
| **ln financial expenditure** | | | | | | 0.997* | 1.273 | -1.419 | 0.0351 | 0.319 |
| | | | | | | (0.542) | (0.903) | (0.973) | (0.297) | (0.389) |
| **ln GDP per capita** | | | | | | -0.24 | -0.0739 | 0.108 | 0.129 | 0.0776 |
| | | | | | | (0.309) | (0.618) | (0.847) | (0.189) | (0.205) |
| **GDP growth rate** | | | | | | -0.0499** | -0.0838 | -0.0381 | 0.00886 | -0.0855* |
| | | | | | | (0.025) | (0.061) | (0.042) | (0.013) | (0.044) |
| **Ratio of secondary industry** | | | | | | 0.000731 | -0.00649 | -0.0387 | -0.00447 | -0.000378 |
| | | | | | | (0.009) | (0.018) | (0.027) | (0.006) | (0.005) |
| **Primary industry ratio** | | | | | | 0.0104 | 0.0363 | 0.0124 | 0.00349 | 0.00117 |
| | | | | | | (0.013) | (0.029) | (0.030) | (0.008) | (0.008) |
| **Sewage treatment rate** | | | | | | -0.000233 | 0.0382 | 0.00202 | 0.00565 | 0.0168 |
| | | | | | | (0.006) | (0.029) | (0.015) | (0.005) | (0.014) |
| **Time effect** | control | control | control | control | control | control | control | control | control | control |
| **Individual effect** | control | control | control | control | control | control | control | control | control | control |
| **Num. obs.** | 655 | 658 | 672 | 664 | 669 | 504 | 510 | 517 | 517 | 510 |
| **Num. groups** | 87 | 87 | 87 | 87 | 87 | 83 | 83 | 83 | 83 | 83 |
| **R-squared** | 0.181 | 0.051 | 0.095 | 0.062 | 0.095 | 0.2344 | 0.0537 | 0.1461 | 0.1365 | 0.1483 |

Note: With standard errors in parentheses.

* $p < 0.05$.

** $p < 0.01$.

*** $p < 0.001$.

public., The procuratorate and the administrative organs are no longer simply the relationship of supervision and supervision, but a partnership that now needs to work together to deal with the assessment of higher authorities to enhance the cohesion of the organization and the power of action.

The reason why the "river chief system + procurator" policy can achieve the expected governance results is also that it goes beyond the institutional framework of the traditional river chief system. The traditional river management system has certain limitations in terms of accountability, and the introduction of the mechanism of the Procurator-General can increase the efforts to hold responsible persons accountable. Procuratorial authorities can investigate, file, and prosecute violations of the law in the course of river governance, ensuring that those responsible perform their duties in accordance with the law and promoting the implementation and effectiveness of governance. For the above reasons, the "river chief system + Procurator General" policy has broken the situation in which the executive and the judiciary act separately in water environmental protection and integrated the government's forces in the water environment and improved the water quality of rivers and lakes on the basis of the effectiveness of the river chief system.

However, further analysis of the results showed that water pollution indicators such as CODMn, DO, PH, and NH3N did not decrease after the implementation of the "river chief

system + procurator" policy, which indicates that this policy has no significant effect on reducing the pollution level of water pollution. We believe that the main reason for these results is the characteristics of the data itself. One reason is that the values of CODMn, DO, PH, NH3N, and other indicators of water pollution are more sensitive as continuous variables, and the detection of data changes significantly. A second reason is that the indicators of water pollution degree are susceptible to precipitation, temperature, light, pollutant discharges, and other uncertainties, and the data related to water quality testing fluctuate frequently. It is difficult to evaluate the actual effect of the implementation of the "river chief system + procurator" policy on the management of water pollutant indicators. In contrast, the water quality class indicator is a categorical variable, and its distribution is relatively stable as an indicator for the comprehensive evaluation of water quality in water bodies.

Therefore, the results of the multiperiod double-difference regression showed that research hypothesis one was true, while research hypothesis two was not. The "river chief system + procurator" policy has performed well in improving the overall water quality of rivers but has performed poorly in reducing the pollution level of rivers. Overall, the intergovernmental cooperation approach to water management has been a relatively successful environmental policy in China, but there is still a need for a larger and deeper intergovernmental cooperation model in the future.

## Robustness test

### Parallel trend test

This paper analyzed the policy effects before and after the implementation of the "river chief system + procurator" in China. To make the conclusions more robust, we used the parallel trend test of the time-varying DID model to test Hypothesis 1 and constructed the following model [19].

$$WQ_{it} = \beta_{\_n}d_{\_n} + \beta_n d_n + \lambda Controls_{it} + \mu_t + \gamma_i + \varepsilon_{it} \tag{2}$$

Table 5 shows the results of the parallel trend test for the above findings, with no control variables added in Column (1) and control variables added in Column (2). Before the implementation of the policy, the regression results were insignificant, and the coefficients were positive. In contrast, after the implementation of the policy, the regression results are significant, and the coefficients are negative. Fig 1 also shows the results of the parallel trend test of the above findings, and the control variables were also added to the test model. As shown in Fig 1, the horizontal coordinates represent the relative time of policy implementation, and the vertical coordinates represent the regression coefficient. The time before the policy implementation is d_5, d_4, d_3, and d_2, and the time after the policy implementation is current, d1, d2, and d3. Before the implementation of the policy, the regression coefficients fluctuate above zero, while after the implementation of the policy, the regression coefficients are all negative. This means that the policy effect of cooperative water ecology management policy among government departments is robust in improving the river water quality and optimizing river water environment. Therefore, the findings of this paper passed the parallel trend test of the time-varying DID method.

### Placebo test

To exclude the interference of omitted variables and other factors, the placebo test in this paper randomizes the treatment group and randomly samples the treatment group variables 500 times, after which it is observed whether the kernel density plots of the randomized DID

**Table 5. Results of the parallel trend test.**

| Time | (1) | (2) |
|---|---|---|
| d_5 | 0.00166 | 0.221 |
| | (0.178) | (0.191) |
| d_4 | 0.0317 | 0.0897 |
| | (0.15) | (0.155) |
| d_3 | 0.0702 | 0.0693 |
| | (0.135) | (0.141) |
| d_2 | 0.138 | 0.119 |
| | (0.124) | (0.128) |
| current | -0.128 | -0.0301 |
| | (0.129) | (0.13) |
| d1 | -0.385** | -0.333* |
| | (0.153) | (0.170) |
| d2 | -0.418** | -0.423** |
| | (0.189) | (0.208) |
| d3 | -0.690** | -0.918** |
| | (0.292) | (0.437) |
| Controls | No | Yes |
| Observations | 570 | 503 |
| R-squared | 0.204 | 0.242 |
| Number of id | 87 | 83 |

Note: With standard errors in parentheses.

* $p < 0.05$.

** $p < 0.01$.

*** $p < 0.001$.

term coefficients and observations are concentrated approximately around 0 and whether the distribution significantly deviates from their true values. As shown in Fig 2, the kernel density plots of the randomized DID term coefficients are concentrated at approximately 0, and the p-value and coefficient distributions are significantly different from the p-value (0.022) and coefficient (-0.327) in the above benchmark regression. In other words, the impact effect in the benchmark analysis is indeed caused by the government's river management policy adjustment, so the findings of this paper are relatively robust.

## Conclusion and discussion

In China, ecological environment management has been given an important position, and the promotion of an ecological civilization has been one of the important contents in the comprehensive construction of a modern socialist country. Faced with the contradiction between frequent outbreaks of water environment crises and people's demand for a good liveable environment, the Chinese government has been exploring more effective water environment policies. Based on the water quality panel data of 87 national water quality monitoring points from the China General Environmental Monitoring Station (GEMS) from 2015 to 2022, this paper empirically examined the impacts of interdepartmental collaboration on water ecosystems under the policy model of the "river chief system + procurator" with the multiperiod double-difference method by using the time of the policy implementation as the explanatory variable and water quality as the explanatory variable.

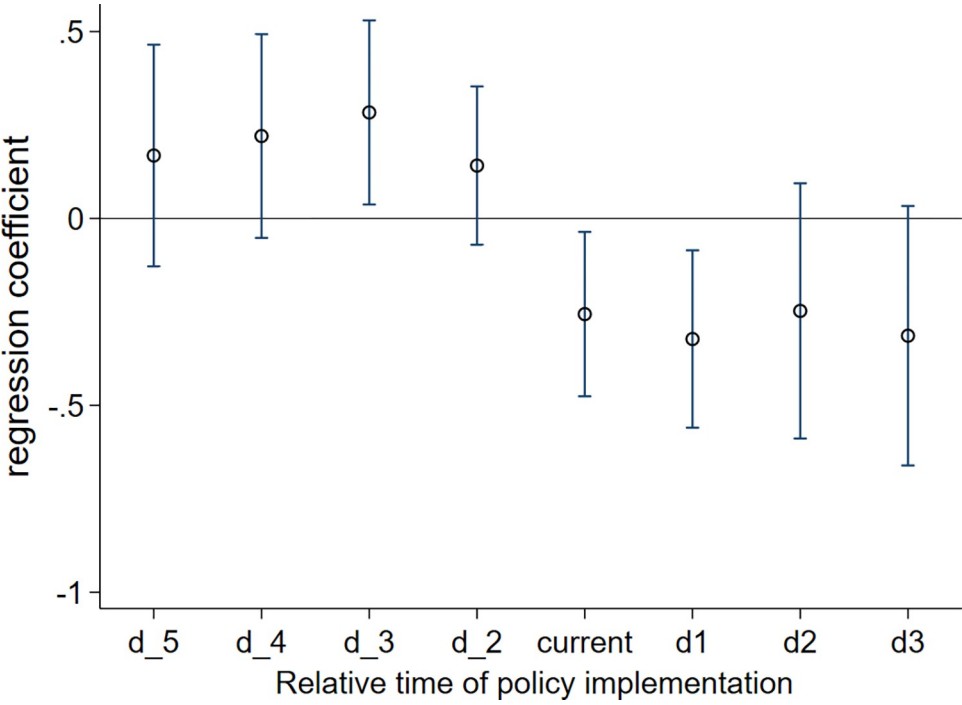

**Fig 1. Results of the parallel trend test.**

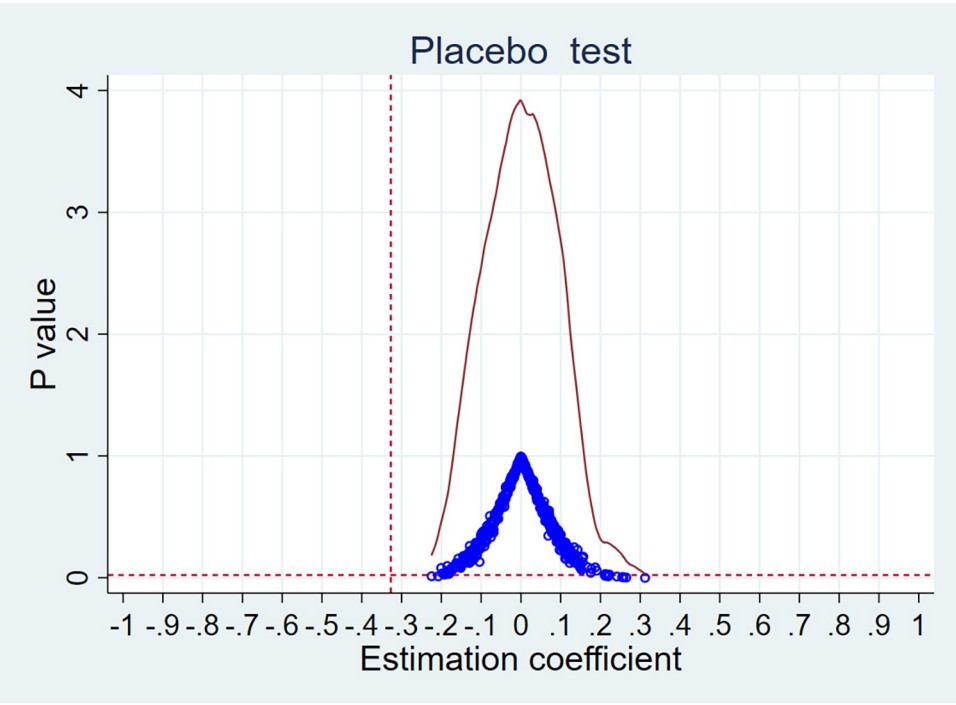

**Fig 2. Placebo test results.**

The data results show that the implementation of the policy of the "river chief + prosecutor system" significantly reduced the water quality class of rivers and lakes and improved their ecological environment, but it was ineffective in reducing the pollutants in rivers and lakes. The study found that the policy of the "river chief system + procurator" integrated decentralized governance forces into a unified institutional framework, which helped solve the problem of fragmentation of water environmental protection rights and responsibilities within the government, promoted the establishment of effective cooperation between river chiefs and procurators, and enhanced the integrity and synergy of water ecological protection.

From the viewpoint of the policy effect of the "river chief system + procurator", river chiefs and procurators have complementary roles in improving river and lake water quality. The river chief can be responsible for monitoring and assessing the water quality of the river and urging the relevant industries and enterprises to strengthen the prevention and control of pollution, while the procurator general can pursue the criminal and administrative liabilities for environmental violations and ensure that the environmental protection actions are carried out smoothly from a legal point of view. The policy-oriented departmental linkage integrates the fragmented administrative resources and becomes the central force in effectively improving the water quality of rivers and lakes. In short, for the environmental management of rivers and lakes, the degree of collaboration among government departments is directly related to the final effect of environmental management.

Finally, as an innovation in the water management system, the policy of the "river chief system + procurator" still has some room for improvement. First, to prevent the government's joint water control policy from becoming a mere formality, this collaborative model of water control should be consolidated in the form of institutionalization and legalization, with a clear system of accountability, rewards, and penalties. Secondly, to improve the effectiveness of the river chief system, it is necessary to explore a wider and deeper interdepartmental cooperative governance mechanism on the basis of the "river chief system + procurator". Public security and courts can be incorporated into the river and lake governance system to promote the organic convergence of administrative law enforcement, criminal justice, procuratorial supervision, and judicial enforcement, to form closed-loop management and whole-process supervision, to crack down on all kinds of crimes and offenses against the environmental resources of rivers and lakes and to provide a long-term and stable guarantee for the maintenance of water quality in rivers and lakes.

## Research contributions and limitations

This paper examines the effect of the implementation of the "river chief system + procurator" policy on the improvement of water quality in rivers and lakes in China and reveals the impact of cooperative policies among government departments on the environmental governance of rivers and lakes, which can help to provide ideas and experience for the country to improve the ecological environment from the perspective of policy formulation. At the same time, there are some limitations in this study. First, the total amount of panel data is insufficient. The implementation of China's "river chief system + procurator" policy was scheduled for approximately 2019, and the policy has been implemented for a relatively short period of time thus far, so the full effect of the policy has yet to be verified. Second, this paper verifies the effects of intergovernmental cooperation policies on river and lake environments within local government jurisdictions and does not include different interregional cooperation within the scope of this paper. However, interregional collaboration is often a key component in consolidating the effects of environmental governance, and future research is expected to discuss and analyze this in greater depth.

## Acknowledgments

We thank the Academic Editor and the reviewers for their useful feedback that improved this paper.

## Author Contributions

**Conceptualization:** Yu Ding.

**Data curation:** Yu Ding, Chen Gong.

**Methodology:** Chen Gong.

**Writing – original draft:** Yu Ding.

**Writing – review & editing:** Chen Gong.

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
