## [Decision Letter · Decision Letter 0]

27 Sep 2023

PONE-D-23-15189Can Intergovernmental Cooperative Policies Promote Water Ecology Improvement - An Analysis Based on Water Quality Data from China's General Environmental Monitoring StationPLOS ONE

Dear Dr. Ding,

Thank you for submitting your manuscript to PLOS ONE. After careful consideration, we feel that it has merit but does not fully meet PLOS ONE’s publication criteria as it currently stands. Therefore, we invite you to submit a revised version of the manuscript that addresses the points raised during the review process.

We look forward to receiving your revised manuscript.

Kind regards,

Chaohai Shen

Academic Editor

PLOS ONE

Journal Requirements:

"This study was supported by the Humanities and Social Sciences Youth Foundation, Ministry of Education of the People's Republic of China (Project No. 22YJC810001)"

Reviewers' comments:

Reviewer's Responses to Questions

**Comments to the Author**

1. Is the manuscript technically sound, and do the data support the conclusions?

Reviewer #1: Yes

Reviewer #2: Partly

2. Has the statistical analysis been performed appropriately and rigorously? 

Reviewer #1: I Don't Know

Reviewer #2: I Don't Know

3. Have the authors made all data underlying the findings in their manuscript fully available?

Reviewer #1: No

Reviewer #2: Yes

4. Is the manuscript presented in an intelligible fashion and written in standard English?

Reviewer #1: Yes

Reviewer #2: No

5. Review Comments to the Author

Reviewer #1: This paper constructed a multi-period differences-in-differences model to evaluate the effectiveness of the cooperative governmental policy of "river chief & procurator" on the improvement of water quality of rivers and lakes based on the data of 87 national water quality monitoring sites released by the China Environmental Monitoring Station from 2015 to 2022. This is a commendable research idea, and its research approach is also basically feasible.However, after reading the entire article, I found that there are still significant areas for improvement.

Therefore, please allow me to provide a major revision and the specific suggestions are as follows:

1.Most intuitively, the format used for the subscripts and references of the model in the text is not standardized. Please make careful modifications.

2.The Hypothesis 1 and Hypothesis 2 seem to overlap to some extent. What is the author's intention in choosing these two hypotheses? This requires a better revelation.

3.The conclusion drawn in the article is slightly weak and there is potential for further exploration.

Reviewer #2: Major revisions are proposed.

This paper studies the latest implementation policy of the “river chief system”, which has strong practical significance. In the process of the implementation of the “river chief system” in China, there are some explorations to enhance the effect of the “river chief system”. In addition, the implementation of “the river chief system” has also improved China's river water environment to some extent.

But the authors' study also has some shortcomings:

1. First of all, in the introduction, the explanation of the mechanism of the “river chief system + procurator” is not sufficient, and the specific content of the system of "river chief + procurator" is not clear. The river chief + Procurator system is mainly produced to deal with the lack of administrative coercive power in dealing with specific river water environment problems, and mainly to enhance the river chief's ability in dealing with water environment problems. Its effect on the management of trans-regional rivers is not within the responsibility of the implementation of the "river chief + prosecutor" policy. Therefore, the conclusions drawn in this paper related to the effect of cross-regional governance need to be further studied.

2. In the 3.0 theoretical mechanism and research hypothesis, the author directly hypothesized the effect of the "river chief + prosecutor system", but the "river chief + prosecutor system" should actually be an extension policy based on the river chief system, and should first study the basis of the river chief system to analyze how the "river chief + prosecutor system" enhances the role of the river chief system. Instead of simply studying the role of the "river chief + procurator system".

3. Line 189 "more than one hundred water quality national control point monitoring stations", should specify the specific number, rather than a general more than one hundred.

4. For line220-225, the selection of control variables seems to be a little arbitrary, and it is hoped that the author can provide specific reasons for the selection of these control variables.

5. The author explains in Line270-281 that the second reason for the river chief system is that cooperation within the government cannot completely solve the problem of fragmented river governance. However, the empirical results in Table 4 fail to see the impact of fragmentation of river governance, so this reason does not seem to be tenable. Whether the river is transregional should be included as a variable in the empirical process for further research.

6. In the discussion part of this paper, the contents of the discussion do not seem to be very consistent with the research results of this paper. The producer should put forward more targeted opinions based on the research of this paper.

6. PLOS authors have the option to publish the peer review history of their article (what does this mean?). If published, this will include your full peer review and any attached files.

Reviewer #1: No

Reviewer #2: No

---

## [Author Response · Author response to Decision Letter 0]

28 Oct 2023

Dear academic editor and reviewer(s)，

 Thanks for your letter and for all reviewer’s comments concerning our manuscript entitled “Can Intergovernmental Cooperative Policies Promote Water Ecology Improvement - An Analysis Based on Water Quality Data from China's General Environmental Monitoring Station”（Manuscript Number PONE-D-23-15189）.

Those comments are all valuable and helpful for revising and improving. We have studied all comments carefully and have made conscientious correction. The revision of the new manuscript includes:①We elaborate on the basis for the research hypothesis②Clarification of the content of the "river chief system + procurator" policy ③Revision of full-text formatting, public notices, tables and data ④Overhauled the concluding section of the paper。⑤Enrichment of some contents.(such as Introduction, Theoretical mechanisms and research hypotheses，Measurements，Conclusion and Discussion）.At the same time, We have professionally touched up and revised the language usage, spelling, and grammar of the article, and have provided the Editing Certificate of manuscript to the academic editor and reviewers. The corrections in the paper and the comments to the reviewer1 and review2 are as flowing.

Reviewer 1

Dear reviewer.

Thanks very much for taking your time to review this manuscript. we really appreciate all your comments and suggestions! Based on your suggestion, we have made corrected modification on the revised manuscript. Furthermore, we would like to show the details as follows:

Comment 1: Most intuitively, the format used for the subscripts and references of the model in the text is not standardized. Please make careful modifications. 

Response:

We thank the reviewer for raising this question. It is true that the model 1 and 2 in the text were not standardized and the model 2 was not sufficiently concise and clear. To address the above issues, we have standardized and modified the models covered in this paper.

Comment 2: The Hypothesis 1 and Hypothesis 2 seem to overlap to some extent. What is the author's intention in choosing these two hypotheses? This requires a better revelation.

Response: 

Thank you for this valuable feedback. We took the issues raised by the reviewers very seriously and thought carefully about the logical relationship between research hypothesis 1 and research hypothesis 2. We believe that although both water quality indicators and water body pollutant indicators are important variables reflecting the water quality of water bodies, there are significant differences between the two. The expression of the research hypotheses in the original manuscript was too simplistic, resulting in a failure to explain clearly the intent of Research Hypothesis 1 and Research Hypothesis 2.

In the new manuscript, we explained the basis for the formulation of Research Hypothesis 1 and Research Hypothesis 2. Specifically, on the one hand, water quality class indicators (Water quality) and water pollutant indicators (PH, DO, CODMn, NH3N) have different roles and importance in evaluating the environmental conditions of rivers and lakes. The water quality class is a composite indicator, which constitutes a reference indicator for the performance assessment of local government officials. China's Ministry of Ecology and Environment issued the "Program for Ranking the Water Environment Quality of National Surface Water Assessment Sections in Prefecture-level and Above Cities (Trial)", which clearly stipulated that the ranking of the water environment quality of each city is determined in accordance with the water quality class. Water pollutants indicators can reflect the sources and categories of pollutants and analyze the shortcomings and deficiencies in the process of water environment management in a more detailed manner.

On the other hand, there are differences in the stability of water quality class indicators (Water quality) and water pollutant indicators (PH, DO, CODMn, NH3N). The indicator of water pollution degree is easily affected by precipitation, temperature, light, pollutant discharge and other uncertainties, so this distinction can be used to examine in more detail which indicator is more significantly affected by the intergovernmental cooperative water management policy of"river chief system + procurator". Therefore, we believe that there is a clear difference between water quality class indicators and water pollutants indicators for measuring the status of river and lake waters, and it is necessary to differentiate between the two.

Comment 3: The conclusion drawn in the article is slightly weak and there is potential for further exploration.

Response:

we appreciate the reviewer for this kind recommendation. We strongly agree with the reviewer's proposal that The conclusion drawn in the article is potential for further exploration. We have rewritten the findings of the new manuscript and the main components include：①Summarizing the article's research findings. The data results show that the implementation of the policy of "river chief system + procurator" can significantly reduce the water quality class of rivers and lakes and improve their ecological environment, but it is ineffective in reducing the pollutants in rivers and lakes. ②Discussion on the effectiveness of the "river chief system + procurator" policy. We believe that the river chief and the procurator-general have complementary roles in improving the water quality of rivers and lakes. The river chief is responsible for monitoring and assessing the water quality of rivers and urging the relevant industries and enterprises to strengthen pollution prevention and control, while the procurator-general is equipped with the deterrent effect of pursuing criminal and administrative liabilities for environmental violations to ensure the successful implementation of environmental protection actions for rivers and lakes. ③Reflecting on the shortcomings of China's "river chief system + procurator" policy and proposing directions for improvement. For example, the collaborative model of combining administrative and judicial forces for water management needs to be institutionalized and legally consolidated. In order to prevent the Government's joint water control policy from becoming a mere formality, a clearer system of accountability, rewards and penalties should also be established.

Thank you very much for your attention and time.

Yours sincerely

Reviewer 2

Dear reviewer.

Thank you very much for your comments and professional advice. These opinions help to improve academic rigor of our article. Based on your suggestion and request, we have made corrected modification on the revised manuscript. Furthermore, we would like to show the details as follows: 

Comment 1:First of all, in the introduction, the explanation of the mechanism of the "river chief system + procurator" is not sufficient, and the specific content of the system of "river chief system + procurator" is not clear. The "river chief system + procurator" is mainly produced to deal with the lack of administrative coercive power in dealing with specific river water environment problems, and mainly to enhance the river chief's ability in dealing with water environment problems. Its effect on the management of trans-regional rivers is not within the responsibility of the implementation of the "river chief system + procurator" policy. Therefore, the conclusions drawn in this paper related to the effect of cross-regional governance need to be further studied. 

Response:

we appreciate the reviewer for this kind recommendation.First of all, we realized the explanation of the mechanism of the "river chief system + procurator" in the introduction part is not sufficient. In the new manuscript, we have strengthened the introduction and explanation of the "river chief system + procurator" in several places. In the introduction section, we introduced the content of the "river chief system + procurator" policy, in detail and emphasized that the river chief system is the basis of the "river chief system + procurator" policy. The essence of the river chief policy lies in centralizing the water environment supervision responsibilities originally scattered among the departments of water conservancy, environmental protection, agriculture, forestry, transportation, fisheries, and oceans into the hands of local party and government leaders. The "river chief system + procurator" is an improvement of the river chief system policy, which expands the scope and form of departmental collaboration on the basis of the river chief system and effectively links administrative law enforcement with criminal justice. At the same time, we have introduced the characteristics of the "river chief system + procurator" system in detail in the Institutional background and characteristic facts section, and we hope that this change will make the contents of the "river chief system + procurator" clearer.

Secondly, we thanks the reviewer for this suggestions on the concluding part of the manuscript. It is true that the "river chief system + procurator" was created primarily to address the lack of administrative coercion in river and lake governance, and had a limited role in influencing trans-regional river management. Therefore, in the new manuscript, we have rewritten the conclusion of the paper, deleted the content of the cross-regional governance effect, and emphasized that the policy of "river chief system + procurator" integrates decentralized sectoral management into the framework of a unified system to a certain extent, and facilitates the establishment of an effective cooperation between the river chiefs and procuratorates, which further enhances the holistic and synergistic nature of water ecology protection.

Finally, we optimized the article by revising the inappropriate expressions throughout the text.

Comment 2: In the 3.0 theoretical mechanism and research hypothesis, the author directly hypothesized the effect of the "river chief + prosecutor system", but the "river chief + prosecutor system" should actually be an extension policy based on the river chief system, and should first study the basis of the river chief system to analyze how the "river chief + prosecutor system" enhances the role of the river chief system. Instead of simply studying the role of the "river chief + procurator system".

Response:

Thank you for this valuable feedback. After discussion we strongly agree with the reviewer's recommendations. We have carefully revised the content of 3.0 Theoretical mechanism and research hypothesis. In the new manuscript, we introduced the fundamental role of river management in the governance of rivers and lakes. The river chief system is to assign the right to control the pollution of the river in each area to the corresponding departmental leaders who will act as "river chiefs"，and be responsible for the management and protection of the corresponding rivers and lakes, so as to solve the problem of fragmentation of the main body of river control. This initiative has contributed to the development of a unified river governance policy, penalty standards and management norms in river governance. On the basis of the river chief system, the "river chief system + procurator" linkage policy is designed to take the convergence of administrative law enforcement with criminal justice and procuratorial public interest litigation as a starting point, and utilize the supervisory power of the judicial sector to urge the government to implement initiatives related to river management in the process of river water quality management. It is hoped that the new manuscript will present more clearly the progressive relationship between the river chief system and the “river chief system + procurator”.

Comment 3: Line 189 "more than one hundred water quality national control point monitoring stations", should specify the specific number, rather than a general more than one hundred.

Response:

we appreciate the reviewer for this kind recommendation. We have rechecked the paper and given the exact data in the text. The monitoring data was recorded for a total of 148 monitoring stations at State control points for water quality from 2007 to 2022.

Comment 4: For line220-225, the selection of control variables seems to be a little arbitrary, and it is hoped that the author can provide specific reasons for the selection of these control variables.

Response:

According to the comments of reviewer, we added specific reasons for the selection of these control variables. We believe that China is in the state of developing countries and the proportion of secondary industry is high, so the environment of rivers and lakes is greatly influenced by human activities and urban economic and social factors. At the same time, we referred to established studies for the selection of control variables, we believe that the overall condition of the regional water environment is closely related to the local level of economic development, industrial structure, financial income and population base. Therefore, the control variables in the regression model to test the effectiveness of the "river chief system + procurator" include economic and social variables such as population, GDP per capita, public revenue, public expenditure, annual growth rate of GDP, proportion of primary industry, proportion of secondary industry, and so on. Considering that the capacity of municipal wastewater treatment may also affect the regional water environment, we will include it as a variable into the empirical model as well. The control variables were taken from the China Urban Statistical Yearbook, and all data were matched to the cities where the monitoring sites were located.

Comment 5:The author explains in Line270-281 that the second reason for the river chief system is that cooperation within the government cannot completely solve the problem of fragmented river governance. However, the empirical results in Table 4 fail to see the impact of fragmentation of river governance, so this reason does not seem to be tenable. Whether the river is transregional should be included as a variable in the empirical process for further research. 

Response: 

We discussed the contents of Line270-281 in detail and concluded that the existing data results really could not illustrate the impact of fragmentation of river governance, so we deleted the contents of Line270-281 and added the reasons for the failure of Research Hypothesis II. We believe that the reasons for the lack of significant changes in the pollution indicators of rivers and lakes after the implementation of the policy of "river chief system + procurator" are as follows: first, As continuous variables, the value of the CODMn, DO, PH, NH3N and other indicators of the water pollution degree is more sensitive and their detection data changes significantly. The second reason is that the indicators of water pollution degree are susceptible to precipitation, temperature, light, pollutant discharges and other uncertainties and the data related to water quality testing will fluctuate frequently. In fact, a number of studies have also mentioned that the external factors affecting water pollution indicators are complex, and we try to explain the regression results of the article more clearly.

Comment 6:In the discussion part of this paper, the contents of the discussion do not seem to be very consistent with the research results of this paper. The producer should put forward more targeted opinions based on the research of this paper.

Response:

we appreciate the reviewer for this kind recommendation. We strongly agree with the reviewer's proposal that the contents of the discussion do not seem to be very consistent with the research results. 

We have rewritten the findings of the new manuscript and the main components include：①Summarizing the article's research findings. The data results show that the implementation of the policy of "river chief system + procurator" can significantly reduce the water quality class of rivers and lakes and improve their ecological environment, but it is ineffective in reducing the pollutants in rivers and lakes. ②Discussion on the effectiveness of the "river chief system + procurator" policy. We believe that the river chief and the procurator-general have complementary roles in improving the water quality of rivers and lakes. The river chief is responsible for monitoring and assessing the water quality of rivers and urging the relevant industries and enterprises to strengthen pollution prevention and control, while the procurator-general is equipped with the deterrent effect of pursuing criminal and administrative liabilities for environmental violations to ensure the successful implementation of environmental protection actions for rivers and lakes. ③Reflecting on the shortcomings of China's "river chief system + procurator" policy and proposing directions for improvement. For example, the collaborative model of combining administrative and judicial forces for water management needs to be institutionalized and legally consolidated. In order to prevent the Government's joint water control policy from becoming a mere formality, a clearer system of accountability, rewards and penalties should also be established.

Thank you very much for your attention and time.

Yours sincerely

---

## [Decision Letter · Decision Letter 1]

13 Nov 2023

Can intergovernmental cooperative policies promote water ecology improvement - An analysis based on water quality data from China's general environmental monitoring station

PONE-D-23-15189R1

Dear Dr. Ding,

We’re pleased to inform you that your manuscript has been judged scientifically suitable for publication and will be formally accepted for publication once it meets all outstanding technical requirements.

Kind regards,

Chaohai Shen

Academic Editor

PLOS ONE

Additional Editor Comments (optional):

Reviewers' comments:

Reviewer's Responses to Questions

**Comments to the Author**

1. If the authors have adequately addressed your comments raised in a previous round of review and you feel that this manuscript is now acceptable for publication, you may indicate that here to bypass the “Comments to the Author” section, enter your conflict of interest statement in the “Confidential to Editor” section, and submit your "Accept" recommendation.

Reviewer #1: All comments have been addressed

Reviewer #2: All comments have been addressed

2. Is the manuscript technically sound, and do the data support the conclusions?

Reviewer #1: Yes

Reviewer #2: Yes

3. Has the statistical analysis been performed appropriately and rigorously? 

Reviewer #1: Yes

Reviewer #2: Yes

4. Have the authors made all data underlying the findings in their manuscript fully available?

Reviewer #1: Yes

Reviewer #2: Yes

5. Is the manuscript presented in an intelligible fashion and written in standard English?

Reviewer #1: Yes

Reviewer #2: Yes

6. Review Comments to the Author

Reviewer #1: (No Response)

Reviewer #2: The author brings the inspection system into the research scope of the river head system and tries to study the influence of the inspection system on the effectiveness of the river head system. The paper excludes some exogenous factors and has a certain degree of innovation.

7. PLOS authors have the option to publish the peer review history of their article (what does this mean?). If published, this will include your full peer review and any attached files.

Reviewer #1: No

Reviewer #2: **Yes: **no

---

## [Editor Report · Acceptance letter]

15 Nov 2023

PONE-D-23-15189R1 

Can intergovernmental cooperative policies promote water ecology improvement - An analysis based on water quality data from China's general environmental monitoring station 

Dear Dr. Ding:

I'm pleased to inform you that your manuscript has been deemed suitable for publication in PLOS ONE. Congratulations! Your manuscript is now with our production department. 

Kind regards, 

on behalf of

Dr. Chaohai Shen 

Academic Editor

PLOS ONE